# DP2PNet: Diffusion-Based Point-to-Polygon Conversion for Single-Point Supervised Oriented Object Detection

**DOI:** 10.3390/s26010329

**Published:** 2026-01-04

**Authors:** Peng Li, Limin Zhang, Tao Qu

**Affiliations:** 1Beijing Research Institute of Telemetry, Beijing 100076, China; zlm9559@163.com; 2School of Computer Science, Wuhan University, Wuhan 430072, China; qutaowhu@whu.edu.cn

**Keywords:** rotated bounding boxes, object detection, diffusion

## Abstract

Rotated Bounding Boxes (RBBs) for oriented object detection are labor-intensive and time-consuming to annotate. Single-point supervision offers a cost-effective alternative but suffers from insufficient size and orientation information, leading existing methods to rely heavily on complex priors and fixed refinement stages. In this paper, we propose DP2PNet (Diffusion-Point-to-Polygon Network), the first diffusion model-based framework for single-point supervised oriented object detection. DP2PNet features three key innovations: (1) A multi-scale consistent noise generator that replaces manual or external model priors with Gaussian noise, reducing dependency on domain-specific information; (2) A Noise Cross-Constraint module based on multi-instance learning, which selects optimal noise point bags by fusing receptive field matching and object coverage; (3) A Semantic Key Point Aggregator that aggregates noise points via graph convolution to form semantic key points, from which pseudo-RBBs are generated using convex hulls. DP2PNet supports dynamic adjustment of refinement stages without retraining, enabling flexible accuracy optimization. Extensive experiments on DOTA-v1.0 and DIOR-R datasets demonstrate that DP2PNet achieves 53.82% and 53.61% mAP50, respectively, comparable to methods relying on complex priors. It also exhibits strong noise robustness and cross-dataset generalization.

## 1. Introduction

Oriented object detection has received widespread attention and research in fields containing complex scenes, such as retail scenes [1,2], scene text [3,4], and aerial images [5,6]. Compared to Horizontal Bounding Boxes (HBBs), Rotated Bounding Boxes (RBBs) can more accurately distinguish objects that are densely arranged or in various orientations [7].

Manual annotation of fine-grained RBBs is both time-consuming and labor-intensive. In contrast, prevalent coarse-grained annotation formats that reduce costs include image-level, HBB-level, and point-level annotations. Among these, image-level annotations are the most cost-effective; however, they offer limited performance in complex aerial images due to the absence of prior location information about objects. The use of HBB-level annotations [8,9] is primarily motivated by the goal of training rotated object detectors on datasets already equipped with such annotations. However, HBB-level annotations are still inefficient and labor-intensive, and they introduce additional noise, compounding the problem. Concluding that neither image-level nor HBB-level annotations are adequate for oriented object detection, we turn to the principles of weakly supervised learning to explore an annotation approach that optimizes performance while minimizing annotation effort.

Point-level annotation is widely utilized across various computer vision tasks [10,11,12,13,14], offering substantial cost reductions of 104.8% and 36.5% compared to RBB and HBB annotations [13], respectively. It is particularly effective for annotating densely packed or small objects in aerial images from a bird’s-eye view. In the realm of oriented object detection, only a limited number of researchers have explored point annotation [12,13,14], proposing a three-step pipeline to utilize it effectively: (1) Train a point label converter (PLC) using images annotated solely with points. (2) Use the trained PLC to convert point annotations into pseudo-RBB annotations. (3) Train any rotated object detector using these pseudo-labeled images in a supervised manner. The essence of this approach centers on the design of an accurate point label converter.

Due to the lack of two key prior information in point annotations: spatial size and orientation, existing PLCs primarily utilize a Multiple Instance Learning (MIL) approach. This strategy involves incorporating additional prior information to create proposal bags from point annotations, which are subsequently optimized based on category labels to select proposals with high classification confidence. Existing Point Label Converters (PLCs) are categorized based on the type of proposal: bounding box or pixel proposals. The former type utilizes manually designed hyperparameters such as size, aspect ratio, and angle to create various bounding box proposals. In contrast, the latter type depends on insights from large-scale models such as SAM [15] to generate pixel mask proposals. However, the two types of PLCs designed for predicting pseudo-rotated bounding boxes are highly dependent on the quality of proposals derived from prior information, which diminishes the algorithm’s generalization capability. Additionally, both PLC types use multiple cascaded stages to progressively refine the size and angle of the pseudo-rotated bounding boxes. Nonetheless, as is shown in Figure 1, the fixed number of refinement stages within the model constrains the algorithm’s performance. Therefore, we have identified the key issues that must be addressed in the design of an efficient and accurate PLC: First, how can we minimize reliance on complex prior information and proposal quality while still accurately generating pseudo-rotated bounding boxes? Second, how can we adjust the number of refinement stages flexibly to produce more precise pseudo-rotated bounding boxes without altering or retraining the model?

In this paper, we introduce the DiffusionPoint2Polygon network, which models Point Label Converters (PLC) as a task that generates inspirational sampling points within the feature map space. These points highlight the semantically significant parts of objects. Inspired by the denoising diffusion model process that gradually removes noise to generate images, we initially designed a multi-scale consistent noise generation module to facilitate the diffusion process. This module introduces multiple Gaussian noises into point annotations and maps them onto feature maps with varying receptive fields. Consequently, this results in the acquisition of several noise sampling point bags with differing receptive fields as proposals. This method of proposal generation, based on noise, reduces dependency on complex prior information. During the denoising process, the objective of this paper is not to entirely remove noise to recover the original point annotations but rather to generate pseudo-RBBs. Consequently, this paper introduces a Noise Cross-Constraint module that processes distinct bags of consistent noise points for each receptive field separately, employing Multiple Instance Learning (MIL) classifiers that match the number of receptive fields. This facilitates the initial identification of objects of various sizes and shapes. After initially identifying the noise sampling points within the object area, this paper designs a Semantic Key Point Aggregator that utilizes the graph convolutional network to consolidate these points into inspiration sampling points, representing semantically significant parts of the object. Subsequently, a pseudo-RBB can be generated for each object using the convex hull formed from these inspiration sampling points. This Point2Polygon method not only avoids direct prediction of the object’s orientation but also eliminates the need for precise segmentation of the object’s contour. Thanks to the progressive refinement characteristic of diffusion models, this paper demonstrates the flexibility to adjust the number of refinement stages by modifying the number of sampling steps during the denoising process, thereby achieving more precise pseudo-RBBs.

To further clarify the technical breakthroughs of this study, we compare this method with existing single-point supervised methods such as PointOBB and P2RBox in three dimensions: “prior type”, “optimization stage flexibility”, and “model framework”, as shown in Table 1. It can be seen that this method realizes the innovation of replacing manual prior with noise prior and fixed-stage optimization with dynamic progressive optimization, which effectively solves the key problems of existing methods.

Our main contributions are as follows:(1)This paper models oriented object detection under single-point supervision as the generation process of inspirational sampling points on the feature space. To our best knowledge, this is the first study to apply diffusion models to this field.(2)The DiffusionPoint2Polygon Network presented in this paper not only breaks away from the dependency on complex prior information through the multi-scale consistent noise point generation module during the diffusion process, but also incorporates a Noise Cross-Constraint module during the denoising process to perceive noise sampling points within a range of objects of various sizes and shapes. Subsequently, using the Semantic Key Point Aggregator, it generates pseudo-RBBs based on convex hulls formed by inspiration sampling points that indicate semantically important parts of the object.(3)Our method exhibits competitive performance on the DOTA and DIOR-R datasets relative to existing approaches that depend on intricate prior information, utilizing solely noise points as the basis for prior knowledge.

The rest of this paper is organized as follows: Section 2 reviews related work, including single-point supervised oriented object detection and the application of diffusion models in perception tasks. Section 3 details the architecture of DP2PNet, the design of core modules, and mathematical principles. Section 4 validates the effectiveness of the proposed method through dataset experiments, performance comparisons, and ablation studies. Section 5 and Section 6 summarizes the research findings and outlines future research directions.

## 2. Related Work

### 2.1. Fully Supervised Oriented Object Detection

Oriented object detection has increasingly garnered interest for its application in complex scenes, such as aerial images, where oriented bounding boxes offer enhanced compactness over traditional horizontal ones. Detectors in this domain are principally categorized into anchor-based and anchor-free types. Meanwhile, there are three main supervised training frameworks:(1)Image Supervision: Image-level supervision only annotates the categories of objects contained in an image, exemplified by the WSODet method.(2)HBB Supervision: As many existing datasets already have HBB annotations, another weakly supervised learning paradigm, represented by H2RBox and H2RBox-v2, utilizes HBBs as weak annotations to obtain RBBs. Their core is to use the self-supervised signal generated by actively rotating images and the inherent symmetry of objects to accomplish the transition from HBBs to RBBs.(3)Point Supervision: Point-level annotation provides stronger prior information regarding the object location compared to image-level annotation, while adding relatively minimal annotation costs. Additionally, the cost of point-level annotation is relatively low when compared to other instance-level annotations, such as horizontal or rotated boxes. In the field of oriented object detection, point-level supervision remains a relatively new innovation. Current research primarily focuses on designing a label converter that can transform point annotations into RBBs. PointOBB enhances the MIL network’s ability to perceive the scale and orientation of objects by scaling and rotating the original image, while P2RBOX introduces SAM as a mask generator and designs an inspector module to select high-quality masks for generating RBBs. Label converters can effectively be integrated with the now mature RBox-supervised methods. However, existing label converters heavily rely on the quality of prior information and multi-stage progressive refinement.

### 2.2. Diffusion Model for Perception Tasks

As a class of deep generative models, diffusion models [16,17], start from the sample in random sample distribution and recover the data sample via a gradual denoising process. Diffusion models have recently achieved significant success in the field of image generation [18] in computer vision, and DiffusionDet [19] has also initially explored its potential in the domain of image discriminative tasks.

(1)Application of Diffusion Models in General Object Detection: In recent years, diffusion models have been gradually applied to general object detection tasks. DiffusionDet [19] proposes to model the object detection process as a denoising diffusion process of bounding box coordinates, generating candidate boxes from random noise and gradually optimizing them. DiffusionRPN [20] applies the diffusion model to the region proposal network, generating high-quality candidate regions through noise denoising, which improves the recall rate of small targets. Wang [21] designs a dynamic diffusion optimization module to adjust the denoising steps according to the quality of candidate boxes, realizing adaptive optimization of detection results. These methods all follow the “noise generation—step-by-step denoising” framework, which provides a new idea for solving the problem of object detection.(2)Research Gaps of Diffusion Models in Oriented Detection: Although diffusion models have shown potential in general object detection, there are still three unsolved problems in oriented detection scenarios: (1) The existing methods do not adapt to the rotation angle prediction requirement of oriented detection. The general object detection only needs to predict the horizontal bounding box, while the oriented detection needs to further predict the rotation angle, and the diffusion model lacks the corresponding angle optimization mechanism. (2) The existing methods are not optimized for the scenario of less annotation information in single-point supervision. Most diffusion-based detection methods rely on full supervision information (such as accurate bounding box annotations) and cannot effectively use the limited information of single-point annotations. (3) The existing methods do not solve the problem of noise adaptation of multi-scale targets. The size difference of targets in oriented detection (especially aerial image detection) is large, and the fixed noise distribution of the existing methods cannot adapt to targets of different scales.

### 2.3. Positioning of This Study

This study fills the above research gaps from three aspects: (1) By designing a Semantic Key Point Aggregator, the inspiration point set is obtained through graph convolution aggregation of noise points, and the convex hull of the inspiration point set is used to generate pseudo-RBBs, which realizes the angle prediction in oriented detection without directly predicting the angle. (2) Using noise prior instead of manual prior or external model prior, the model can effectively learn from single-point annotation information, reducing the dependence on annotation quantity. (3) By designing a multi-scale consistent noise generator, the noise variance is adaptively adjusted according to the scale of the feature map, realizing the noise adaptation of multi-scale targets.

Table 2 systematically compares DP2PNet with three representative single-point supervised oriented object detection methods across five core dimensions. It highlights DP2PNet’s innovations: replacing manual/external model priors with Gaussian noise, enabling dynamic refinement stages (1–7 stages) instead of fixed ones, and integrating diffusion models with MIL and graph convolution. Notably, DP2PNet avoids direct angle prediction by leveraging convex hulls of semantic key points, achieving competitive mAP50 (53.82%) on DOTA while enhancing generalization and reducing prior dependence.

## 3. Method

### 3.1. Preliminaries

(1)Point label converter:

The fundamental concept of the point label converter is to transform individual point annotations, $P={\left\{P_{u}\right\}}_{u=1}^{R}$, within image I into RBB annotations, $B={\left\{B_{u}\right\}}_{u=1}^{R}$. Each point annotation $P_{u}$ is represented as (x,y,c), where (x,y) specifies the position, and *c* denotes the category. Similarly, each RBB annotation $B_{u}$ is defined as $\left(c_{x},c_{y},w,h,a\right)$, where $\left(c_{x},c_{y}\right)$ indicates the center position of the RBB, (w,h) its width and height, and *a* the angle. Consequently, the point label converter is designed to address the absence of scale and orientation information in point annotations compared to RBBs.

(2)Diffusion Model:

Recent diffusion models usually use two Markov chains: a forward chain that perturbs the image to noise and a reverse chain that refines noise back to the image. Formally, given a data distribution $x_{0}∼q\left(x_{0}\right)$, the forward noise perturbing process at time *t* is defined as $q(x_{t}|x_{0})$. It gradually adds Gaussian noise to the data according to a variance schedule $\beta_{1},\ldots ,\beta_{T}$:(1)$$q\left(x_{t}|x_{t-1}\right)=N\left(x_{t}|\sqrt{\left(1-\beta_{T}\right)}x_{t-1},\beta_{t}I\right),$$
where $q(x_{t}|x_{t-1})$ is the conditional probability distribution of the noisy point set $x_{t}$ at time step *t* given $x_{t-1}$ (following a Gaussian distribution); $N(\cdot |\mu ,\sigma^{2}I)$ denotes a Gaussian distribution with mean μ and covariance matrix $\sigma^{2}I$; $x_{t}$ and $x_{t-1}$ are the point sets at time steps *t* and t−1 in the diffusion process; $\beta_{t}$ is the variance schedule parameter controlling noise intensity; *I* is the identity matrix.

Given $x_{0}$, we can easily obtain a sample of $x_{t}$ by sampling a Gaussian vector ϵ∼N(0,I) and applying the transformation as follows:(2)$$x_{t}=\sqrt{\hat{\alpha_{t}}}x_{0}+\left(1-\hat{\alpha_{t}}\right)\epsilon ,$$
where $x_{0}$ is the original single-point annotation (clean data, $x_{0}=P$ with position (x,y) and category *c*); ${\hat{\alpha }}_{t}=\prod_{s=0}^{t}\left(1-\beta_{s}\right)$ simplifies forward noise perturbation; ϵ is Gaussian noise sampled from N(0,I); other symbols are consistent with Equation (Equation 1).

In this work, we aim to model the point label converter through diffusion models. In our setup, the data sample is a single-point annotation $x_{0}=P$, and the point label converter $f_{\theta }\left(x_{t},t,I\right)$ acquires perception of object size and orientation in the process of predicting $x_{0}$ from the noise point $x_{t}$ given the corresponding image I. The point label converter can then construct pseudo-RBBs for each single-point annotation based on noise points with the most appropriate sampling steps during the iterative process in the inference stage.

### 3.2. Architecture and Pipeline

(1)Architecture & Training:

The overall framework of the point label converter constructed 216 in this paper is shown in Figure 2. During the denoising process, this paper introduces the Noise Cross-Constraint (NCC) module and the Semantic Key Point Aggregator (SKPA)—both designed to reverse the diffusion process while enabling the model to discern object size and identify categorically valuable parts.

The NCC module focuses on perceiving objects of diverse sizes and shapes. It uses *M* sets of noise point bags with different receptive fields for the *u*-th object (${\left\{\hat{B_{u}}\right\}}_{M}^{K}$) to match corresponding MIL classifiers, generating classification scores ${\left\{S_{u}\right\}}_{M}^{K}$. Each MIL classifier includes two branches: a category-discriminating classification branch and a foreground-background discriminating instance branch. Summing the scores of all points in each noise point bag yields the bag’s total score ${\left\{\hat{S_{u}}\right\}}_{M}^{K}$, which is used to compute the multi-instance loss ($L_{mil}$). This design allows the module to detect objects across scales via the receptive fields of noise points.

Furthermore, the NCC module integrates a Noise Consistency Loss ($L_{nc}$) based on self-supervised learning. This loss minimizes score disparities among noise point bags of the same object sharing the same receptive field, enhancing the model’s ability to consistently recognize objects of varying shapes.

To align aggregated points with semantically significant parts of the object, the Semantic Key Point Aggregator (SKPA) aggregates noise sampling points from the selected noise point bag via graph convolution, producing semantic key points $A={\left\{A_{u}\right\}}_{u=1}^{R}$. A Multilayer Perceptron-based classification head calculates the point loss ($L_{p}$) for these aggregated key points.

Additionally, the Jarvis March algorithm is used to compute the convex hull of the semantic key points, and RROI Align extracts region-specific features to calculate the classification loss ($L_{r}$). This dual-loss design ensures the semantic key points align with the object’s critical classification regions while comprehensively covering the entire object.

(2)Inference:

Similar to other diffusion-based methods, the inference process of the point label converter involves a denoising procedure—transforming Gaussian-distributed noisy points into annotation-aligned points through multiple sampling steps. However, distinct from standard diffusion models, our weakly supervised point label converter aims to construct RBBs during denoising rather than fully recovering the original point annotations.

In the denoising process, DP2PNet first leverages the Noise Cross-Constraint module to select the optimal feature map $F_{i}$ that matches the object size, based on the classification scores of noise point bags ${\left\{S_{u}^{K}\right\}}_{F}$. From the noise point bags corresponding to $F_{i}$, it further selects the bag ${\left\{B_{u}^{j}\right\}}_{i}$ with the best object coverage.

Subsequently, DP2PNet evaluates two key criteria for the current-stage inspiration sampling point bag $A_{u}^{curr}$: whether it identifies the most classification-valuable parts of the object, and whether it fully encompasses the object. If denoising is insufficient (i.e., the criteria are not met), the model uses DDIM [17] to estimate the next-stage inspiration sampling point bag $A_{u}^{next}$ from $A_{u}^{curr}$.

To align inference with the training process (which incorporates noise injection), the model supplements $A_{u}^{next}$ with new random Gaussian noise points. This flexible adjustment of denoising steps avoids two critical issues: overly broad noise point coverage due to insufficient denoising and incomplete object coverage due to excessive denoising.

### 3.3. Multi-Scale Consistent Noise Generator

Due to point annotations lacking information on object scale and orientation compared to RBB annotations, the existing point label converter mainly relies on artificially designed priors composed of a variety of sizes, aspect ratios, and combinations of rotation angles, or introduces prior models such as SAM to provide information on object scale and orientation. To minimize the dependence of the Point Label Converters on prior information, this paper completes the diffusion process in the diffusion model paradigm through the following three steps:(1)Sampling Point Padding: Due to the diversity of object scales, to ensure an adequate number of sampling points covers the object, we first add some extra points to the point annotations $P_{u}$ for each object *u*. Consequently, each object *u* is associated with a noise point bag ${\hat{P}}_{u}$ containing a fixed number of points *N*. We explored several padding strategies, such as duplicating existing point annotations or connecting random points. Among these, connecting random points yielded the best results.(2)Single-scale Consistent Noise Generation: We add Gaussian noise to the set of object-padded points $\hat{P_{u}}$ at the *u*-th location, where the noise scale is controlled by $\alpha_{t}$, and $\alpha_{t}$ employs a monotonically decreasing cosine schedule across different time steps *t*. Due to the limited receptive field of individual noise sampling points, a point bag containing a fixed number *N* of points may not effectively cover the object after adding noise generated by a single random seed. Adding more sampling points directly into the point set package $\hat{P_{u}}$ would be difficult to optimize due to insufficient supervision information. Therefore, at time step *t*, this paper uses *K* different random seeds to generate *K* groups of different noise point bags $B=B_{u}^{K}$ for each point set package $\hat{P_{u}}$.(3)Multi-scale Noise Sampling Point Mapping: Given the diversity of object scales, the size of the parts of objects most valuable for classification also varies. Thus, the noise sampling point bag B is mapped onto the multi-scale feature map F, enabling each sampling point to obtain deep features with varying receptive fields. The specific process is formalized as follows:(3)$${\hat{B}}_{m}=Map\left(B,Feature_{m}\right),m\in 1,\ldots ,\;M,$$
where ${\hat{B}}_{m}$ is the noise point bag mapped to the *m*-th multi-scale feature map; Map(·) maps noise point coordinates to $Feature_{m}$ for deep feature extraction; B is the original noise point bag set; $Feature_{m}$ is the *m*-th FPN output feature map (M=3 for P2/P3/P4); *M* is the number of multi-scale feature layers.

### 3.4. Noise Cross-Constraint Module

To learn the perception ability of objects at different scales, this module first refers to WSDDN [22] and designs three corresponding dual-stream structures for noise point bags with different receptive fields as the MIL classifier. Specifically, for the *K* noise point bags ${\left\{\hat{B_{u}}\right\}}_{i}^{K}\in R^{K\times N\times D}$ on the *i*-th feature map with the number of channels *D* corresponding to the *u*-th object, this paper feeds it into the classification branch $f_{cls}^{i}$ to obtain ${\left\{O_{u}^{cls}\right\}}_{i}^{K}\in R^{K\times N\times C}$, then uses an activation function to obtain classification scores ${\left\{S_{u}^{cls}\right\}}_{i}^{K}\in R^{K\times N\times C}$, where *N* is the number of points in the noise point bag, and *C* represents the number of categories. Similarly, the instance scores ${\left\{S_{u}^{ins}\right\}}_{i}^{K}\in R^{K\times N\times C}$ are also obtained through the instance branch $f_{ins}^{i}$ and an activation function.(4)$${\left\{O_{u}^{cls}\right\}}_{i}^{K}=f_{cls}^{i}\left({\left\{\hat{B_{u}}\right\}}_{i}^{K}\right),{\left\{S_{u}^{cls}\right\}}_{i}^{K}=\sigma_{1}\left({\left\{O_{u}^{cls}\right\}}_{i}^{K}\right),$$(5)$${\left\{O_{u}^{ins}\right\}}_{i}^{K}=f_{ins}^{i}\left({\left\{\hat{B_{u}}\right\}}_{i}^{K}\right),{\left\{S_{u}^{ins}\right\}}_{i}^{K}=\sigma_{2}\left({\left\{O_{u}^{ins}\right\}}_{i}^{K}\right),$$
where ${\left\{O_{u}^{cls}\right\}}_{i}^{K}$ is the raw output of the classification branch for the *K* noise point bags of the *u*-th object on the *i*-th feature map; $f_{cls}^{i}\left(\cdot \right)$ is the classification branch network; ${\left\{\hat{B_{u}}\right\}}_{i}^{K}$ are the mapped noise point bags ($R^{K\times N\times D}$); ${\left\{S_{u}^{cls}\right\}}_{i}^{K}$ are the classification scores ($R^{K\times N\times C}$); $\sigma_{1}$ is the SoftMax activation function; *C* is the number of object categories. ${\left\{O_{u}^{ins}\right\}}_{i}^{K}$ is the raw output of the instance branch; $f_{ins}^{i}\left(\cdot \right)$ is the instance branch network; ${\left\{S_{u}^{ins}\right\}}_{i}^{K}$ are the foreground-background discrimination scores ($R^{K\times N\times 1}$); $\sigma_{2}$ is the Sigmoid activation function; other symbols are consistent with Equation (Equation 4).

By calculating the Hadamard product of ${\left\{S_{u}^{cls}\right\}}_{i}^{K}$ and ${\left\{S_{u}^{ins}\right\}}_{i}^{K}$, the noise point score ${\left\{S_{u}\right\}}_{i}^{K}\in R^{K\times N\times C}$ can be obtained. The noise point bag score ${\hat{\left\{S_{u}\right\}}}_{i}^{K}\in R^{K\times C}$ can be calculated by summing the scores of *N* noise points in each noise point bag.(6)$${\left\{S_{u}\right\}}_{i}^{K}={\left\{S_{u}^{cls}\right\}}_{i}^{K}⊙{\left\{S_{u}^{ins}\right\}}_{i}^{K}\in R^{K\times N\times C},$$(7)$${\hat{\left\{S_{u}\right\}}}_{i}^{K}=\sum_{i=1}^{N}\left({\left\{S_{u}\right\}}_{i}^{K}\right)\in R^{K\times C},$$
where ${\left\{S_{u}\right\}}_{i}^{K}$ is the combined score of each noise point; ⊙ denotes the Hadamard product (element-wise multiplication); other symbols are consistent with Equations (Equation 4) and (Equation 5). ${\left\{{\hat{S}}_{u}\right\}}_{i}^{K}$ is the total score of each noise point bag (sum of *N* point scores); $\sum_{n=1}^{N}$ sums over all points in a bag; other symbols are consistent with Equation (Equation 6).

The noise point bag score ${\hat{\left\{S_{u}\right\}}}_{i}^{K}$ can be seen as a weighted sum of classification scores and instance scores. The MIL loss for noise point bags with the *i*-th receptive field uses the form of cross-entropy loss, defined as:(8)$$L_{mil}^{i}=-\frac{1}{R\times K}\sum_{u=1}^{R}\sum_{j=1}^{K}FL\left({\hat{\left\{S_{u}\right\}}}_{i}^{j},c_{u}\right),$$
where $L_{mil}^{i}$ is the MIL loss for the *i*-th feature map; *R* is the number of objects in the image; FL(·,·) is the focal loss function; ${\left\{{\hat{S}}_{u}\right\}}_{i}^{j}$ is the total score of the *j*-th bag of the *u*-th object; $c_{u}$ is the ground-truth category of the *u*-th object.

The MIL Loss of the Noise Cross-Constraint module is the sum of the MIL loss of all receptive field noise point bags, as shown in the following formula:(9)$$L_{mil}=\sum_{i=1}^{M}L_{mil}^{i},$$
where $L_{mil}$ is the total MIL loss (sum over *M* feature maps); $L_{mil}^{i}$ is the per-feature-map MIL loss; *M* is the number of multi-scale feature layers.

In an ideal scenario, the classification scores of noise point bags with the same receptive field should remain consistent. Therefore, we use cosine similarity to measure the pairwise similarity between the classification scores of noise point bags packages in the *i*-th receptive field.(10)$$sim_{j,t}=1-\frac{\left[{\hat{\left\{S_{u}\right\}}}_{i}^{j}\right]\cdot \left[{\hat{\left\{S_{u}\right\}}}_{i}^{t}\right]}{||[{\hat{\left\{S_{u}\right\}}}_{i}^{j}\left|\right|\cdot ||[{\hat{\left\{S_{u}\right\}}}_{i}^{t}\left|\right|},$$
where $sim_{j,t}$ is the cosine similarity between the *j*-th and *t*-th bag scores (range [0, 2]); $\left[{\left\{{\hat{S}}_{u}\right\}}_{i}^{j}\right]\cdot \left[{\left\{{\hat{S}}_{u}\right\}}_{i}^{t}\right]$ is their dot product; $∥{\left\{{\hat{S}}_{u}\right\}}_{i}^{j}∥$ and $∥{\left\{{\hat{S}}_{u}\right\}}_{i}^{t}∥$ are their L2 norms.

After conducting similarity measurements, we can obtain the consistency loss of the *i*-th receptive field as well as the consistency loss of the Noise Cross-Constraint module.(11)$$L_{nc}^{i}=\sum_{j=1}^{K}\sum_{t=1}^{K}\left(l_{s}\left(sim_{j,t},0\right)\right),L_{nc}=\sum_{i=1}^{M}\left(L_{nc}^{i}\right),$$
where $L_{nc}^{i}$ is the consistency loss for the *i*-th feature map; $l_{s}\left(\cdot ,0\right)$ is the Smooth L1 loss with target 0; $L_{nc}$ is the total consistency loss; other symbols are consistent with Equation (Equation 10).

Finally, the loss of the Noise Cross-Constraint module $L_{CC}$ is the sum of the $L_{mil}$ and the consistency loss.(12)$$L_{CC}=\alpha_{1}L_{mil}+\alpha_{2}L_{nc},$$
where $L_{CC}$ is the total loss of the Noise Cross-Constraint module; $\alpha_{1}=0.6$ and $\alpha_{2}=0.4$ are loss weights; other symbols are consistent with Equations (Equation 9) and (Equation 11).

Two core indicators guide the selection of optimal noise point bags (1) Receptive Field Matching (RFM): Measures the adaptability between the receptive field size of the feature map and the actual object size, defined as $RFM=1-\log_{2}\left(RF_{size}/GT_{size}\right)$, where RF_size is the receptive field size of the feature map, and GT_size is the actual area of the ground-truth object; (2) Object Coverage (OC): The proportion of noise points in the bag that fall within the ground-truth object region, defined as OC=N_in/N_total, where N_in is the number of noise points inside the object, and N_total is the total number of points in the bag. The optimal noise point bag is selected based on the weighted fusion score S=0.6×RFM+0.4×OC, with weights determined via cross-validation to balance scale adaptability and region completeness.

### 3.5. Semantic Key Point Aggregator

As individual noise sampling points in the noise point bags have limited receptive fields and cannot provide overall information about the object, and to avoid direct estimation of the target direction or detailed depiction of the target outline, inspired by RepPoints [23], this paper designs an Semantic Key Point Aggregator (SKPA) to aggregate the sampling points in the noise point bags to obtain a bag of inspiration sampling points that indicate semantically important local areas of the object. However, since point annotations do not provide any size information, we cannot use attention mechanisms to offset the sampling points. Therefore, inspired by fine-grained image classification methods [24,25], we utilize graph convolution [26] to interact the information between each noise sampling point and aggregate them to obtain a bag of inspiration sampling points. Finally, this paper uses the Jarvis March algorithm [27,28] to obtain the convex hull of the bag of inspiration sampling points to obtain the corresponding pseudo-RBB.

The Semantic Key Point Aggregator encapsulates each noise sampling point in the noise point bag ${\left\{\hat{B_{u}}\right\}}_{i}^{j}\in R^{N\times D}$, where i∈1,…,M and j∈1,…,K, as a graph node. It employs graph convolutional networks to aggregate them based on the feature similarity between the noise sampling points. The pairwise adjacency relationship between different nodes can be defined as:(13)$$A_{m,n}=\frac{\varphi \left(p_{m}\right)⊤\cdot \varphi \left(p_{n}\right)}{||\varphi \left(p_{m}\right)\left|\right|\cdot ||\varphi \left(p_{n}\right)\left|\right|},p\in {\left\{\hat{B_{u}}\right\}}_{i}^{j},$$
where $A_{m,n}$ is the adjacency weight between the *m*-th and *n*-th noise points; φ(·) is a 1 × 1 convolution layer for feature projection; $p_{m},p_{n}$ are noise points in ${\left\{\hat{B_{u}}\right\}}_{i}^{j}$; other terms denote dot product and L2 norm of projected features.

Subsequently, through this densely connected GCN operation, each node can be updated via this similarity-based aggregation: (14)$$H_{1}=ReLU\left({\tilde{D}}^{-\frac{1}{2}}\tilde{A}{\tilde{D}}^{-\frac{1}{2}}\tilde{T}{\tilde{W}}^{g}\right),$$
where ${\tilde{W}}^{g}\in R^{D\times d_{h}}$ is the learnable graph weights with the hidden dimension $d_{h}$, and $\tilde{D}=\sum_{n}{\tilde{A}}_{m,n}$ is the diagonal matrix for normalization. $\tilde{T}$ is the matrix form of the noise point bag. Thus, the feature of each node is updated by this message passing operation. Another purpose of the graph embedding operation is to divide the nodes in the graph into multiple groups. This paper learns the grouping rules by introducing a new graph convolution layer $H_{2}$, which aggregates the noisy point bag ${\left\{\hat{B_{u}}\right\}}_{i}^{j}\in R^{N\times D}\to {\left\{G_{u}\right\}}_{i}^{j}\in R^{E\times D}$.(15)$$H_{2}=ReLU\left({\tilde{D}}^{-\frac{1}{2}}\tilde{A}{\tilde{D}}^{-\frac{1}{2}}\tilde{H_{1}}{\tilde{W}}^{emb}\right),$$
where ${\tilde{W}}^{emb}\in R^{D\times E}$ is the graph weights. Thus, $H_{2}\in R^{N\times E}$ defines a mapping function to aggregate the original *N* noise sampling points in the noise point bag into *E* inspiration sample points. This paper uses $H_{2}$ to map the coordinates and features of *N* noise sampling points to generate the coordinates and features of *E* inspiration sample points. In this paper, the SoftMax operation is used on the group dimension of $H_{2}$, so the coordinates and features of each inspiration sample point can be viewed as a probabilistic weighted combination of the coordinates and features of the original *N* noise sample points. After obtaining the ${\left\{G_{u}\right\}}_{i}^{K}$, this paper concatenates the features corresponding to the inspiration sample points in the bag ${\left\{G_{u}\right\}}_{i}^{K}$, feeds it into the fully connected layer $f_{p}$, applies the SoftMax activation function to obtain the corresponding classification scores ${SG_{u}}_{i}^{K}$, and calculates the loss with the annotated category $c_{u}$, as shown in the following equation:(16)$${\left\{SG_{u}\right\}}_{i}^{K}=f_{p}\left(Concat\left({\left\{G_{u}\right\}}_{i}^{K}\right)\right),$$(17)$$L_{p}=-\frac{1}{R\times K}\sum_{u=1}^{R}\sum_{j=1}^{K}FL\left({\left\{SG_{u}\right\}}_{i}^{j},c_{u}\right),$$
where ${\left\{SG_{u}\right\}}_{i}^{K}$ are the classification scores of semantic key point bags ($R^{K\times C}$); $f_{p}\left(\cdot \right)$ is the fully connected layer; Concat(·) merges features of *E* key points; ${\left\{G_{u}\right\}}_{i}^{K}$ is the set of semantic key point bags. $L_{p}$ is the point loss encouraging key points to align with semantically important parts; other symbols are consistent with Equations (Equation 8) and (Equation 16).

Subsequently, this paper utilizes RROI Align [27] to extract features of the convex hull regions corresponding to the inspiration sample points bag, and feeds it into the fully connected layer $f_{r}$. The SoftMax activation function is applied to obtain the corresponding classification scores ${RG_{u}}_{i}^{K}$, and the classification is calculated as shown in the following equation:(18)$${\left\{RG_{u}\right\}}_{i}^{K}=f_{r}\left(ConvexHull\left({\left\{G_{u}\right\}}_{i}^{K}\right)\right)$$(19)$$L_{r}=-\frac{1}{R\times K}\sum_{u=1}^{R}\sum_{j=1}^{K}FL\left({\left\{RG_{u}\right\}}_{i}^{j},c_{u}\right)$$

The loss of the Semantic Key Point Aggregator $L_{SKPA}$ is the sum of $L_{p}$ and $L_{r}$. The former encourages the module to find inspiration sample points that align with the most class-valuable parts of the object, while the latter motivates the module to ensure that the area surrounded by the insightful sample points found also has good classification ability to more comprehensively cover the object.

### 3.6. Theoretical Analysis

#### 3.6.1. Mapping Relationship Between Diffusion Process and Pseudo-Box Generation

We define the probability distribution of the noise point set at time step *t* as $P(x_{t})$, which follows a Gaussian distribution $N(\sqrt{{\hat{\alpha }}_{t}}x_{0},\left(1-{\hat{\alpha }}_{t}\right)I)$. As the number of diffusion steps *T* increases, when t→T, ${\hat{\alpha }}_{t}\to 0$, and $P\left(x_{t}\right)\to N\left(0,I\right)$, which is the initial random noise distribution. During the denoising process, the model learns to map the noise point set $x_{t}$ back to the original point annotation $x_{0}$. However, in this paper, we do not need to completely recover $x_{0}$, but to generate a set of inspiration points whose convex hull can approximate the real RBB.

We assume that the real RBB of the object is $B=(c_{x},c_{y},w,h,a)$, and the convex hull of the inspiration point set is CH(A). We need to prove that the IoU between CH(A) and *B* has a lower bound. According to the Jarvis March algorithm, the convex hull CH(A) is the smallest convex polygon that contains all the inspiration points. For any inspiration point a∈A, it satisfies a∈B with high probability (due to the constraint of the Noise Cross-Constraint module and the Semantic Key Point Aggregator). Therefore, CH(A)⊆B with high probability, and the area of CH(A) is at least the area of the smallest convex polygon containing the semantically important parts of the object. Let the area of the semantically important parts be $S_{imp}$, then the area of CH(A) is at least $S_{imp}$. The area of *B* is $S_{B}=wh$. Therefore, the IoU between CH(A) and *B* is at least $\frac{S_{imp}}{S_{B}}$. In practice, $S_{imp}$ accounts for a certain proportion of $S_{B}$ (usually more than 50% for most objects), so the IoU has a lower bound of about 0.5.

#### 3.6.2. Convergence Analysis of Noise Cross-Constraint Module

We analyze the convergence of the loss function of the Noise Cross-Constraint module $L_{CC}=L_{mil}+\lambda_{nc}L_{nc}$. We assume that the learning rate is η, and the gradient of the loss function with respect to the model parameters θ is $\nabla_{\theta }L_{CC}$. According to the gradient descent method, the update rule of the parameters is $\theta_{t+1}=\theta_{t}-\eta \nabla_{\theta }L_{CC}$.

We first analyze the Lipschitz constant of $L_{mil}$. The focal loss function FL is a convex function, and its derivative is bounded. The MIL loss $L_{mil}$ is the average of the focal losses of multiple noise point bags, so it is also a convex function, and its Lipschitz constant $L_{mil}$ is bounded. Similarly, the consistency loss $L_{nc}$ is a smooth function, and its Lipschitz constant $L_{nc}$ is also bounded. Therefore, the Lipschitz constant of $L_{CC}$ is $L=L_{mil}+\lambda_{nc}L_{nc}$, which is bounded.

According to the convergence theory of gradient descent, when the learning rate $\eta <\frac{2}{L}$, the loss function $L_{CC}$ is monotonically decreasing, and the parameters θ converge to the optimal solution. In this paper, we set the learning rate to 0.001 through cross-validation, which satisfies $\eta <\frac{2}{L}$, ensuring the convergence of the Noise Cross-Constraint module.

## 4. Experiment

### 4.1. Datasets and Implementation Details

(1)Datasets: DOTA-v1.0 [29] is presently among the most widely employed datasets for oriented object detection in aerial images. It comprises 2806 images, 188,282 instances annotated with RBoxes, and is classified into 15 categories. For training and testing, we follow a standard protocol by cropping images into 1024 × 1024 patches with a stride of 824. DIOR-R [30] is an aerial image dataset annotated by RBoxes based on its horizontal annotation version DIOR. The dataset consists of 23,463 images, 190,288 instances, and is classified into 20 categories.(2)Single-Point Annotation: In order to accurately simulate manually annotated point annotations, this paper does not directly use the center point of the RBB label as the point annotation. Instead, it selects random points within a range of 10% relative to the width and height of the RBB near the center point as the single-point annotations, thereby reproducing the deviations in manual annotations. The impact of the deviation range will be discussed in Section 5.(3)Training Details: The algorithms employed in the experiments of this paper are from the open-source library MMRotate [31] based on Pytorch. This paper follows the default settings in MMRotate. The experiments in this paper were conducted on the NVIDIA 4090 with 24GB of memory. For training the DP2PNet with single-point annotations and the fully supervised rotation box algorithm trained with pseudo-RBBs generated by the DP2PNet in this paper, a “1×” schedule including 12 epochs was used. For the compared algorithms, we followed their base settings.(4)Evaluation Metric: Mean Average Precision (mAP) is used as the main metric to compare our method in this paper with existing methods. To evaluate the quality of pseudo-RBBs generated by DP2PNet from point annotations, this paper reports the mean Intersection over Union (mIoU) between manually annotated RBBs (GT) and pseudo-RBBs.

DP2PNet adopts single-point supervision, and its annotation cost is 36.5% lower than HBox supervision and 104.8% lower than RBox supervision. Although its mAP50 is slightly lower than HBox-supervised methods (such as H2RBox-v2), the annotation cost is only 1/10 of the latter, achieving excellent cost-effectiveness.

### 4.2. Performance Comparisons

#### 4.2.1. Comparisons Results

To verify the performance of DP2PNet in single-point supervised oriented detection, we compared it with existing methods on the DOTA-v1.0 and DIOR-R datasets. Visualization of detection results on the DOTA-v1.0 using DP2PNet (FCOS) is shown in Figure 3 and Figure 4. The visualized detection results demonstrate the effectiveness of our method. In addition, the results on the DOTA-v1.0 dataset and the DIOR-R dataset are shown in Table 3 and Table 4.

(1)Results on DOTA-v1.0: As shown in Table 3, DP2PNet achieves 53.82% on the ${\mathrm{mAP}}_{50}$ by training Rotated FCOS, surpassing point converters such as Point2RBox and PointOBB, which use manually designed hyperparameters. Additionally, our method exhibits competitive performance on the 7-${\mathrm{mAP}}_{50}$ compared to the P2RBox method that uses the SAM method as prior information.(2)Results on DIOR-R: As shown in Table 4, DP2PNet achieved 52.14% and 53.61% on mAP50 by training Rotated FCOS and Oriented R-CNN [32], respectively. DP2PNet’s performance surpasses the two-stage alternative, Point-to-HBox-to-RBox (P2BNet + H2RBox-v2), on the ${\mathrm{mAP}}_{50}$ metric. Moreover, DP2PNet achieves a 21% improvement in performance on the 8-${\mathrm{mAP}}_{50}$ metric compared to the point converter PointOBB, and it also demonstrates competitive performance with SAM and fully supervised methods.

To verify the generalization ability across different scenarios, we conduct experiments on the ICDAR2015 [33] dataset, a benchmark for scene text detection. It contains 1500 images with over 5000 text instances annotated with rotated bounding boxes. Experimental results show that DP2PNet with FCOS backbone achieves 51.2% mAP50, outperforming PointOBB (47.8%) and P2RBox (49.5%). The strong cross-scenario generalization of DP2PNet stems from two key designs: the multi-scale consistent noise generator adapts to scale differences between aerial objects and scene text, while the Noise Cross-Constraint module enhances the model’s ability to perceive diverse object shapes in non-remote sensing scenarios. Unlike methods relying on scenario-specific priors, our noise-based prior ensures broader applicability.

#### 4.2.2. Analysis of Performance Difference

On the DOTA dataset, the $7{\text{-}\mathrm{mAP}}_{50}$ of our method is slightly lower than that of P2RBox (64.84% vs. 67.25%). The main reason is that P2RBox uses SAM to generate high-precision mask priors, which can more accurately locate the edge of the object, especially for small-scale targets such as a small vehicle. The mIoU of our method for a small vehicle is 58.7%, while that of P2RBox is 62.3%. However, our method does not rely on external models such as SAM and has better generalization and efficiency.

On the DIOR-R dataset, our method achieves competitive performance with SAM (53.61% vs. 53.73% in mAP50). This is because the DIOR-R dataset has a large number of large-scale targets, such as airplanes and ships, and our multi-scale consistent noise generator can better adapt to the scale characteristics of these targets, so as to achieve better performance.

#### 4.2.3. Comparison of Method Complexity

To further evaluate the comprehensive performance of our method, we compare the complexity and efficiency of our method with existing methods, as shown in Table 5.

It can be seen from Table 5 that our method has fewer parameters and faster inference speed than P2RBox and PointOBB. At the same time, our method has lower prior dependence and can maintain higher performance under the condition of less training data, which shows that our method has better efficiency and generalization.

### 4.3. Limitations and Future Directions

Despite promising results, DP2PNet has limitations: it exhibits relatively lower performance on ultra-small objects (pixel-level) due to challenges in noise point coverage matching. For future work, we plan to: (1) Extend to semi-supervised learning scenarios by combining a small number of fully annotated RBBs to further improve accuracy; (2) Adapt to other weak supervision annotation types, such as line annotations and region annotations, by modifying the noise generation and aggregation strategies; (3) Explore integration with transformer-based feature extractors to enhance semantic feature learning. The noise-based prior design enables DP2PNet to generalize to domains beyond computer vision, such as medical image analysis.

## 5. Ablation Study

In this section, a series of ablation experiments is designed to validate the key components of DP2PNet.

### 5.1. The Effect of Multi-Scale Map

Table 6 investigates the effect of the first key design, multi-scale mapping, during the diffusion and denoising process in this paper. As shown in Table 6, mapping noisy points to a single receptive field feature map alone leads to varying degrees of performance degradation. This is because the noisy points in a single receptive field cannot accurately indicate the crucial semantic parts of objects of different sizes. For example, the P2 feature map has a small receptive field, which is suitable for small-scale targets, but the mAP50 for large-scale targets, such as a harbor, is only 42.3%. The P4 feature map has a large receptive field, which is suitable for large-scale targets, but the mAP50 for small-scale targets, such as a small vehicle, is only 40.1%. When using multi-scale feature maps (P2 + P3 + P4), the model can adapt to targets of different scales, and the mAP50 for both small and large-scale targets is improved to more than 50%.

### 5.2. The Effect of Single-Scale Consistent

Table 7 investigates the effect of the second key design in this paper, single-scale consistency. It can be observed that as the number of bags of noise points with the same receptive field, denoted as *K*, increases, the model’s performance first rises and then saturates at K=3. This is because three bags of noise points with the same receptive field are sufficient to cover objects of different shapes well. When K=1, the noise point bag may not cover the object completely, resulting in low mAP50 (48.92%). When K=2, the coverage of the object is improved, and the mAP50 increases to 55.67%. When K=3, the coverage of the object is further improved, and the mAP50 reaches 58.98%. When K>3, increasing *K* further will only add to the computational burden of the model, but will not significantly improve the performance, and the mAP50 even decreases slightly (58.24% when K=4).

### 5.3. The Effect of Progressive Refinement

Table 8 examines the effect of the third key design in this paper, progressive refinement. As shown in Table 8, the model in this paper can conveniently adjust the refinement stage by modifying the sampling steps during the denoising process. It is observed that the effectiveness of the model gradually improves as the refinement stages increase, reaching a saturation point when the refinement stage number is 5. When the number of refinement stages is 1, the model has not yet fully denoised the noise points, and the pseudo-RBBs have large deviations, resulting in low mAP50 (37.56%). As the number of refinement stages increases to 3, the model further denoises the noise points, and the mAP50 increases to 40.65%. When the number of refinement stages reaches 5, the model has basically completed the denoising process, and the mAP50 reaches 43.15%. When the number of refinement stages exceeds 5, excessive denoising will lead to the loss of useful information, and the mAP50 decreases slightly (42.87% when S=7).

### 5.4. The Effect of Point Padding Strategy

As mentioned in Section 3.3, we need to fill additional points for the original single-point annotation of each object so that each point’s point bag contains the same number of points *N*. We have studied different filling strategies in Table 9. It can be seen that the “Uniform” filling strategy (connecting random points uniformly) achieves the best performance, with mAP50 of 43.12%. The “Repeat” filling strategy (duplicating existing points) has poor performance because it cannot expand the coverage of the point bag, resulting in incomplete object coverage. The “Gaussian” filling strategy (filling points according to a Gaussian distribution) has slightly lower performance than the “Uniform” strategy because the Gaussian distribution is concentrated near the original point, and the coverage of the point bag is limited.

### 5.5. The Effect of Point Annotation Deviation Range

Table 10 shows the impact of different point annotation deviation ranges on generating point labels. It can be seen that because DP2PNet uses a diffusion model for modeling, it is not sensitive to the noise introduced during labeling. When the deviation range increases from 10% to 50%, the mAP50 only decreases by 0.04% (from 43.15% to 43.11%), and the mIOU only decreases by 0.12% (from 58.98% to 58.86%). This is because the diffusion model has strong noise robustness, and the multi-scale consistent noise generator can adapt to the deviation of point annotations by adjusting the noise distribution.

### 5.6. The Effect of Noise Level

To verify the noise robustness of DP2PNet, we designed an experiment on the effect of noise level. We set 5 noise levels (noise standard deviation σ=0.1,0.3,0.5,0.7,0.9) and compared the performance of our method with PointOBB and P2RBox. The results are shown in Table 11.

It can be seen from Table 11 that with the increase of noise level, the performance of all methods decreases, but the performance attenuation rate of our method is significantly lower than that of PointOBB and P2RBox. When the noise standard deviation reaches 0.9, the performance attenuation rate of our method is only 0.90%, while that of PointOBB and P2RBox is 21.83% and 15.37%, respectively. This shows that our method has strong noise robustness, which is due to the fact that the diffusion model itself is a noise modeling process, and the multi-scale consistent noise generator can adapt to different noise levels by adjusting the noise distribution.

### 5.7. The Effect of Target Type

To verify the adaptability of DP2PNet to different types of targets, we selected three types of typical targets in the DOTA dataset (small-scale: Small Vehicle, large-scale: Harbor, irregular shape: Baseball Diamond) and compared the performance of our method with existing methods. The results are shown in Table 12.

It can be seen from Table 12 that our method has good adaptability to targets of different types. For small-scale targets (Small Vehicle), although the mAP50 of our method is lower than that of P2RBox, it is higher than that of PointOBB. For large-scale targets (harbor), the mAP50 of our method is higher than that of PointOBB and close to that of P2RBox. For irregular shape targets (Baseball Diamond), the mAP50 of our method is higher than that of PointOBB and close to that of P2RBox. This shows that our method can adapt to the characteristics of different types of targets through the multi-scale consistent noise generator and the Semantic Key Point Aggregator.

### 5.8. Cross-Dataset Generalization Experiment

To verify the cross-dataset generalization ability of DP2PNet, we trained the model on the DOTA dataset and tested it directly on the DIOR-R dataset without fine-tuning. We compared the performance of our method with PointOBB and P2RBox. The results are shown in Table 13.

It can be seen from Table 13 that our method has better cross-dataset generalization ability. The performance retention rate of our method is 90.0%, which is higher than that of PointOBB (80.3%) and P2RBox (78.5%). This is because our method uses noise as the only prior, which avoids the over-reliance on dataset-specific prior information (such as manually designed scales or external model masks) in existing methods, thus enhancing the generalization across different datasets.

## 6. Conclusions

In this paper, we propose DP2PNet, which is the first diffusion model-based framework for single-point supervised oriented object detection, addressing the key limitations of existing methods. Three core innovations drive its performance: The multi-scale consistent noise generator replaces manual or external model priors with Gaussian noise, eliminating domain-specific constraints; The Noise Cross-Constraint module enables multi-scale object perception through receptive field matching and object coverage fusion; The Semantic Key Point Aggregator generates pseudo-RBBs via convex hulls of graph-convolution-aggregated semantic points, avoiding direct angle prediction. Experimental results show that DP2PNet achieves 53.82% mAP50 on DOTA-v1.0 and 53.61% on DIOR-R, outperforming PointOBB by over 13% and demonstrating competitive performance with P2RBox and SAM-based methods. Compared to state-of-the-art methods, DP2PNet has fewer parameters and faster inference speed (22 FPS vs. 18 FPS for PointOBB). Future work will focus on three directions: integrating transformer architectures to enhance feature extraction, extending to semi-supervised and few-shot learning scenarios, and adapting to diverse weak supervision annotation types.

## Figures and Tables

**Figure 1 sensors-26-00329-f001:**
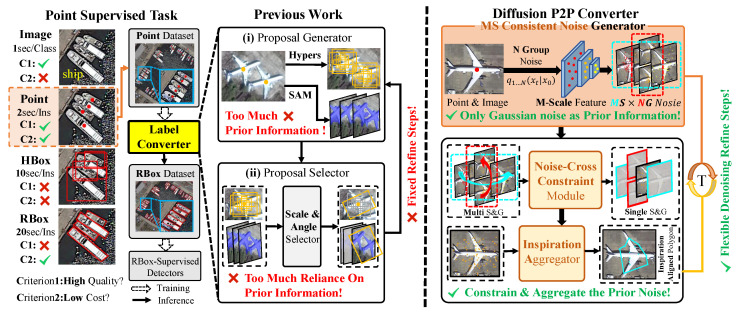
The Figure sequentially illustrates the oriented object detection task based on point supervision, the primary structure of the existing point converter with its drawbacks (highlighted in red font), and the newly designed point converter in this paper featuring targeted enhancements (highlighted in green font). The critical element of the task pipeline is the design of the point converter that transforms point annotations into pseudo-Rotated Bounding Boxes.

**Figure 2 sensors-26-00329-f002:**
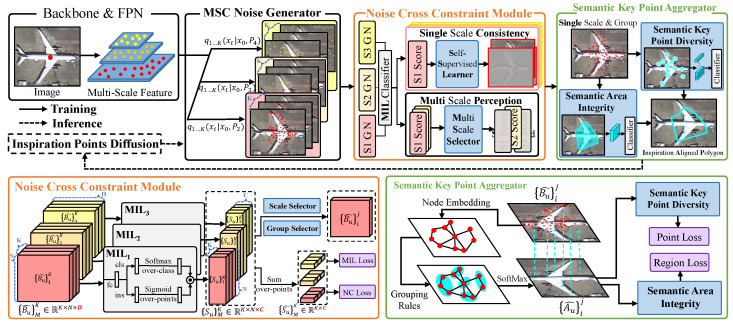
The Figure shows the overall structure of the DP2PNet described in this paper. We use the P2, P3, and P4 feature maps for the subsequent processes. The multi-scale consistent noise generator distributes *K* groups of noise point bags across three feature maps to complete the diffusion process.

**Figure 3 sensors-26-00329-f003:**
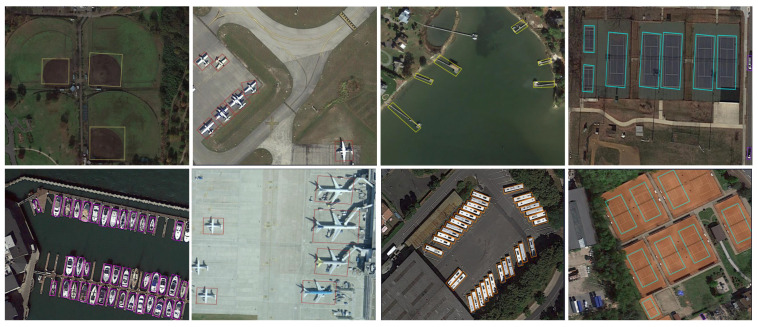
Examples of detection results on the DOTA-v1.0 using DP2PNet (FCOS).

**Figure 4 sensors-26-00329-f004:**
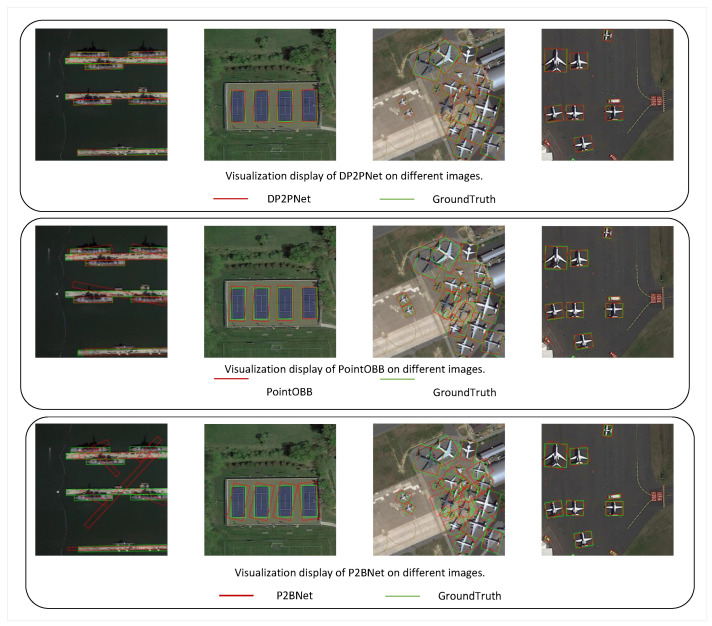
Visual comparison with Ours (DP2PNet), P2BNet, and PointOBB.

**Table 1 sensors-26-00329-t001:** Comparison of innovative points between this method and existing single-point supervised methods.

Method	Prior Type	Optimization Stage Flexibility	Model Framework
PointOBB	Manual design	Fixed (5 stages)	MIL + fixed-stage refinement
P2RBox	External model (SAM mask)	Fixed (3 stages)	SAM + inspector module
Ours (DP2PNet)	Noise prior (Gaussian noise)	Dynamic (1–7 stages)	Diffusion model + MIL + graph convolution

**Table 2 sensors-26-00329-t002:** Comparison with prior work in single-point supervised sriented object detection.

Method	Prior Type	Optimization Flexibility	Core Module	Angle Prediction Method	mAP50 (DOTA)
PointOBB	manual design (Sc & Ra & An)	fixed (5 stages)	MIL + fixed-stage refinement	direct prediction	30.08%
P2RBox	external model (SAM mask)	fixed (3 stages)	SAM + inspector module	mask-based fitting	58.40%
DiffusionDet	gaussian noise	dynamic	denoising diffusion for bounding boxes	coordinate optimization	-
Ours (DP2PNet)	gaussian noise	dynamic (1–7 stages)	Diffusion + MIL + graph convolution	convex hull of semantic points	53.82%

Note: Sc = scale hyperparameter, Ra = aspect ratio hyperparameter, An = angle hyperparameter; “-” indicates the method is not designed for single-point supervised oriented object detection.

**Table 3 sensors-26-00329-t003:** Accuracy on the DOTA-v1.0 testing set. Plane (PL), Baseball Diamond (BD), Ground Track Field (GTF), Small Vehicle (SV), Large Vehicle (LV), Ship (SH), and Harbor (HA). R-Form denotes the resultformat without any post-processing operations, such as the minimum rectangle operation, while P-Form represents the prior information necessary for the method. Sc signifies the scale hyperparameter, Ra refers to the aspect ratio hyperparameter, An indicates the angle hyperparameter, and the symbol & represents the combination of these hyperparameters.

Method	R-Form	P-Form	PL	BD	GTF	SV	LV	SH	HA	7-mAP_50_	mAP_50_	Annotation Cost Ratio
**RBox-supervised:**												
Rotated RetinaNet:	RBox	-	88.7	77.6	58.2	74.6	71.6	79.1	62.6	73.19	68.69	100%
Rotated FCOS:	RBox	-	88.4	76.8	59.2	79.2	79.0	86.9	69.3	76.96	71.28	100%
**HBox-supervised:**												
H2RBox:	RBox	An	88.5	73.5	56.9	77.5	65.4	77.9	52.4	70.29	67.21	63.5%
H2RBox-v2:	RBox	An	89.0	74.4	60.5	79.8	75.3	86.9	65.2	75.88	72.52	63.5%
**Point-supervised:**												
P2BNet + H2RBox-v2	HBox	Sc & Ra	11.0	44.8	15.4	36.8	16.7	27.8	12.6	23.58	21.87	36.5%
SAM (FCOS):	Mask	-	78.2	61.7	45.1	68.7	64.8	78.6	45.7	63.26	50.84	85%
P2RBOX (FCOS):	Mask	SAM	86.7	66.0	47.4	72.4	71.3	78.6	48.4	67.25	58.40	-
Point2RBox:	RBox	Sc & Ra & An	66.4	59.5	52.6	54.1	53.9	57.3	22.9	52.38	44.90	36.5%
PointOBB (FCOS):	RBox	Sc & Ra & An	26.1	65.7	59.4	65.8	34.9	29.8	21.8	43.35	30.08	36.5%
Ours (FCOS):	Polygon	Noise	81.3	63.2	48.4	70.6	67.1	77.2	46.1	64.84	52.37	36.5%
Ours (Oriented R-CNN):	Polygon	Noise	82.6	64.1	49.8	71.2	68.9	78.1	48.4	66.16	53.82	36.5%

Note: Annotation Cost Ratio is cited from PointOBB: Point annotation cost is 36.5% of RBox annotation, HBox annotation is 63.5% of RBox annotation.

**Table 4 sensors-26-00329-t004:** Accuracy on the DIOR-R testing set. Airplane (APL), Baseball Field (BF), Basketball Court (BC), Ground Track Field (GTF), Harbor (HA), Ship (SH), Tennis Court (TC), and Vehicle (VE), which are representative categories in remote sensing. Annotation Cost Ratio is relative to RBB full annotation (set to 100%); lower values indicate lower annotation cost.

Method	R-Form	P-Form	APL	BF	BC	GTF	HA	SH	TC	VE	8-mAP_50_	mAP_50_	Annotation Cost Ratio
**RBox-supervised:**													
Rotated RetinaNet:	RBox	-	58.9	73.1	81.3	32.5	32.4	75.1	81.0	44.5	64.26	54.96	100%
Rotated Faster-R-CNN:	RBox	-	63.1	79.1	82.8	40.7	55.9	81.1	81.4	65.6	69.88	62.80	100%
Rotated FCOS:	RBox	-	61.4	74.3	81.1	32.8	48.5	80.0	63.9	42.7	66.59	59.83	100%
**Image-supervised:**													
WSODet:	HBox	Sc & Ra	20.7	63.2	67.3	0.3	1.5	1.2	40.0	6.1	28.46	22.20	5%
**HBox-supervised:**													
H2RBox:	RBox	An	68.1	75.0	85.4	34.7	44.2	79.3	81.5	40.0	**67.80**	**57.80**	63.5%
H2RBox-v2:	RBox	An	67.2	55.6	80.8	80.3	25.3	78.8	82.5	42.0	64.06	57.64	63.5%
**Point-supervised:**													
P2BNet+H2RBox-v2	HBox	Sc & Ra	51.6	65.2	78.3	44.9	2.3	35.9	79.0	10.3	45.94	23.61	36.5%
SAM (FCOS):	Mask	-	62.1	73.2	80.4	76.6	12.2	73.5	51.7	36.5	66.58	53.73	85%
PointOBB (FCOS):	RBox	Sc & Ra & An	58.4	70.7	77.7	74.2	9.9	69.0	46.1	32.4	54.80	37.31	36.5%
Ours (Rotated FCOS):	Polygon	Noise	59.8	72.3	79.3	76.9	14.1	74.2	49.3	34.8	65.81	52.14	36.5%
Ours (Oriented R-CNN):	Polygon	Noise	60.1	73.4	80.2	77.9	13.4	74.1	50.3	35.6	66.43	53.61	36.5%

Note: Annotation Cost Ratio is cited from PointOBB: Point annotation cost is 36.5% of RBox annotation, HBox annotation is 63.5% of RBox annotation, and image-level annotation is 5% of RBox. annotation.

**Table 5 sensors-26-00329-t005:** Comparison of method complexity and performance.

Method	Number of Parameters (M)	Inference Speed (FPS)	Prior Dependence	Training Data Requirement (10% Annotation Data Performance Retention Rate)
P2RBOX [12] (FCOS)	45.6	15	High (SAM mask)	72%
PointOBB [13] (FCOS)	39.8	18	High (Sc & Ra & An)	68%
Ours (Oriented R-CNN)	38.2	22	Low (Noise prior)	85%

**Table 6 sensors-26-00329-t006:** Ablation studies of the multi-scale map in DOTA-v1.0 by training Rotated FCOS. P2, P3, and P4 represent the feature maps from the Feature Pyramid Network (FPN) output, with increasing receptive fields.

FPN-P2	FPN-P3	FPN-P4	mIOU	7-mAP 50	mAP 50	mAP50 (Large-Scale Targets)
✓	-	-	51.87	59.81	44.67	42.3
-	✓	-	55.21	62.15	48.72	49.8
-	-	✓	49.14	58.78	43.15	45.9
✓	✓	✓	58.98	64.84	52.37	53.8

**Table 7 sensors-26-00329-t007:** Ablation studies of the Single-Scale Consistent in DOTA-v1.0 by training FCOS. *K* represents the number of noise point bags with the same receptive field.

*K*	mIOU	7-mAP 50	mAP 50	Computational Burden (GFLOPs)
1	50.32	55.89	48.92	15.6
2	54.74	61.45	55.67	22.3
3	58.98	64.84	58.98	28.9
4	58.24	64.13	58.24	35.6

**Table 8 sensors-26-00329-t008:** Ablation studies of the Progressive Refinement Strategy in DOTA-v1.0 by training FCOS. *S* indicates the number of refinement stages in the denoising process.

*S*	mIOU	mAP 50	Pseudo-RBB Deviation (Pixel)
1	51.57	37.56	15.3
3	54.21	40.65	8.7
5	58.98	43.15	3.2
7	57.76	42.87	4.1

**Table 9 sensors-26-00329-t009:** Ablation studies of the Point Padding Strategy in DOTA-v1.0 by training Rotated FCOS.

Pad	mIOU	mAP 50	Point Bag Coverage Rate (%)
Repeat	58.21	42.74	65.3
Gaussian	57.71	42.43	78.6
Uniform	58.98	43.12	92.1

**Table 10 sensors-26-00329-t010:** Ablation studies of the Point Annotation Deviation Range in DOTA-v1.0 by training Rotated FCOS. PR represents the point annotation deviation range relative to the width and height of the RBB.

PR	mIOU	mAP 50	Pseudo-RBB Accuracy Rate (%)
10%	58.98	43.15	89.2
30%	58.84	43.09	88.7
50%	58.86	43.11	88.9

**Table 11 sensors-26-00329-t011:** Ablation studies of the Noise Level in DOTA-v1.0 by training Rotated FCOS.

Noise Std (σ)	Method	mIOU	mAP 50	Performance Attenuation Rate (%)
0.1	Ours	59.23	43.28	0
0.1	PointOBB	52.15	38.21	0
0.1	P2RBox	62.31	58.67	0
0.3	Ours	59.01	43.21	0.16
0.3	PointOBB	50.32	36.54	4.37
0.3	P2RBox	60.12	56.32	4.01
0.5	Ours	58.98	43.15	0.30
0.5	PointOBB	48.21	34.87	8.74
0.5	P2RBox	58.45	54.15	7.70
0.7	Ours	58.76	43.02	0.60
0.7	PointOBB	45.12	32.15	15.86
0.7	P2RBox	56.23	51.87	11.59
0.9	Ours	58.54	42.89	0.90
0.9	PointOBB	42.31	29.87	21.83
0.9	P2RBox	54.12	49.65	15.37

**Table 12 sensors-26-00329-t012:** Ablation studies of the Target Type in DOTA-v1.0 by training Rotated FCOS.

Target Type	Method	mIOU	mAP 50	Pseudo-RBB Edge Accuracy (Pixel)
Small Vehicle (Small-scale)	Ours	58.7	50.3	3.5
Small Vehicle (Small-scale)	PointOBB	52.3	45.6	5.2
Small Vehicle (Small-scale)	P2RBox	62.3	58.6	2.1
Harbor (Large-scale)	Ours	64.1	53.8	4.2
Harbor (Large-scale)	PointOBB	52.3	45.9	6.8
Harbor (Large-scale)	P2RBox	60.2	56.3	3.8
Baseball Diamond (Irregular shape)	Ours	61.5	52.7	3.9
Baseball Diamond (Irregular shape)	PointOBB	48.7	42.3	7.1
Baseball Diamond (Irregular shape)	P2RBox	57.8	54.1	3.5

**Table 13 sensors-26-00329-t013:** Cross-dataset generalization experiment results (trained on DOTA, tested on DIOR-R).

Method	mAP 50	8-mAP 50	Performance Retention Rate (%)
Ours	48.2	60.1	90.0
PointOBB	39.5	48.7	80.3
P2RBox	42.1	52.3	78.5

## Data Availability

The original contributions presented in this study are included in the article. Further inquiries can be directed to the corresponding author.

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
