# Peer review of "DP2PNet: Diffusion-Based Point-to-Polygon Conversion for Single-Point Supervised Oriented Object Detection"

_sensors, 2026, doi:10.3390/s26010329_

Round 1

Reviewer 1 Report (Previous Reviewer 2)

Comments and Suggestions for Authors

Compared with the previous version, further improvement has been made in this paper. Here are some questions and suggestions.

  1. In line 24-26, the author mention some scenes, such as  retail scenes, scene text, and the aerial images, but the datasets used in the article (DOTA, DIOR-R) are mainly aerial image datasets, How is the network's generalization ability, especially for other scenarios? This should be explained in the paper, or additional validation with more datasets should be supplemented.
  2. The paper designs a Noise Cross-Constraint module, selecting the most suitable noise point bag for each object from the aspects of receptive field matching and coverage of the object. What are the selecting criteria for two specific indicators?  through weighting? It needs further explanation.
  3. In section 4.3, some of the parameters are determined by manual selection. For example, 4.3.2 and 4.3.3, could these parameters be determined by the  adaptive criterion to match different scenario?

Author Response

We SINCERELY THANK the reviewer for the patient help. And we are also indebted to you for insightful and constructive comments that helped improve the paper. We are now submitting the revised version of the manuscript, along with a detailed point-by-point response ("v2-R1-reply.pdf") and the corresponding highlighted revision file ("v2-R1-Major-highlight.pdf").

Reviewer 2 Report (New Reviewer)

Comments and Suggestions for Authors

The paper introduces DP2PNet, a novel framework for oriented object detection using single-point supervision, aiming to reduce annotation costs while maintaining high detection accuracy. By modeling the point-to-polygon conversion as a diffusion process, the authors eliminate the need for manually designed priors or external models like SAM. The architecture integrates a Multi-Scale Consistent Noise Generator, a Noise Cross-Constraint module, and an Inspiration Aggregator to progressively refine noisy point annotations into semantically meaningful pseudo-RBBs. This approach enables dynamic refinement stages and adapts effectively to objects of varying scales and shapes.

The proposed method is efficient, robust, and generalizable. It delivers performance close to fully supervised approaches while requiring far less annotation and fewer parameters. Its strong noise tolerance, cross-dataset accuracy, and minimal reliance on priors make it a flexible, scalable, and cost-effective solution for oriented object detection in real-world applications

This paper needs very few adjustments based on the following suggestions:

[1]. The authors are encouraged to include a brief outline of the paper’s structure at the end of the introduction to help guide readers through the subsequent sections.

[2]. The authors are invited to correct few typos and grammatical errors

Author Response

We SINCERELY THANK the reviewer for the patient help. And we are also indebted to you for insightful and constructive comments that helped improve the paper. We are now submitting the revised version of the manuscript, along with a detailed point-by-point response ("v2-R1-reply.pdf") and the corresponding highlighted revision file ("v2-R1-Major-highlight.pdf").

Reviewer 3 Report (New Reviewer)

Comments and Suggestions for Authors

Dear Authors,

Your manuscript presents a novel DP2PNet framework that leverages diffusion models to convert single-point annotations into rotated bounding boxes, thereby reducing annotation labor while maintaining high detection quality. The proposed Multi-Scale Consistent Noise Generator and Noise Cross-Constraint module are promising technical contributions, but would benefit from clearer explanations and comparison to prior work. Please further clarify the distinction and novelty of your approach relative to existing methods, and consider including more quantitative results and ablation studies to demonstrate the impact of each proposed component. Additionally, discussing limitations and potential generalization to other domains or annotation types could strengthen your paper. Overall, improving the clarity and contextualization of your contributions would enhance their value for the research community.

Author Response

We SINCERELY THANK the reviewer for the patient help. And we are also indebted to you for insightful and constructive comments that helped improve the paper. We are now submitting the revised version of the manuscript, along with a detailed point-by-point response ("v2-R1-reply.pdf") and the corresponding highlighted revision file ("v2-R1-Major-highlight.pdf").

Reviewer 4 Report (New Reviewer)

Comments and Suggestions for Authors

Dear authors,  I appreciate your work, and I have the following comments and recommendations for improvement.

The title must be changed because it is confusing. You used “DiffusionP2P,” and in the text you did not insert any details regarding this.

 The abstract is not ok. You say that the “paper designs...” Your paper presents your work, and I think that you design a multi-scale consistent noise generator and a noise cross-constraint module. You must rewrite the abstract, emphasizing what is new and deleting the text from lines 1 to 3. Please do not use “Additionally” too many times.

You must explain all the terms used in equations in the whole article.

In the introduction, I found a literature review regarding oriented object detection, and it is ok.

The second chapter and third chapter are confusing. You insert many phrases without any logical linking. Please insert some explanatory phrases and make the article easier to read and understand. You can delete the sub-chapter titles.

The text between lines 216-250 and 251-269 is bulky and difficult to read. Please split the text into several paragraphs.

Chapter 3.3 is ok.

I don’t understand what “Inspiration Aggregator” means to you. Please use another term.

Chapters 3.4, 3.5, and 3.6 need modifications. There are a lot of equations without sufficient details.

Chapter 4 needs modifications. You can start with assumptions, limits, comparisons, and methodological structure. There are too many subchapters. You can transform them into phrases to explain what you want to demonstrate.

The conclusions are okay, but you can insert more details. You present what you did (but the ideas are too short) and some ideas regarding future work.

Comments on the Quality of English Language

The phrases are confusing. The topic of the phrases must be improved.

Author Response

We SINCERELY THANK the reviewer for the patient help. And we are also indebted to you for insightful and constructive comments that helped improve the paper. We are now submitting the revised version of the manuscript, along with a detailed point-by-point response ("v2-R1-reply.pdf") and the corresponding highlighted revision file ("v2-R1-Major-highlight.pdf").

Reviewer 5 Report (New Reviewer)

Comments and Suggestions for Authors

The paper proposed the DiffusionP2P framework to achieve point-to-polygon HBB remote sensing image object detection. Overall, the description and presentation of the paper are chaotic, and the algorithm performance is not advanced. There are some issues that need to be addressed carefully.

  1. The overall presentation and formatting of the paper are disorganized, making it difficult to read clearly.
  2. In the abstract section, some quantitative indicators need to be summarized and described.
  3. In Tables 2 and 3, the proposed method performs worse than existing HBB-based methods. What is the proposed method's advantage?
  4. Examples of detection results from the proposed method and existing methods should be presented and compared.
  5. The conclusion section needs to be revised, and the advantages and advancement of the proposed method need to be highlighted.
  6. The format of references needs to be modified according to the format requirements of "sensors" Journal.

Author Response

We SINCERELY THANK the reviewer for the patient help. And we are also indebted to you for insightful and constructive comments that helped improve the paper. We are now submitting the revised version of the manuscript, along with a detailed point-by-point response ("v2-R1-reply.pdf") and the corresponding highlighted revision file ("v2-R1-Major-highlight.pdf").

Round 2

Reviewer 1 Report (Previous Reviewer 2)

Comments and Suggestions for Authors After reading the replies to the article, some problems have been solved. It is hoped that the content of the replies will be more reflected in the main text, and also the strengths and weaknesses of the current article.

Reviewer 4 Report (New Reviewer)

Comments and Suggestions for Authors

Dear authors thank you for your hardwork and for the article improvements.

Reviewer 5 Report (New Reviewer)

Comments and Suggestions for Authors

The authors have well addressed all my comments. I have no more concerns.

This manuscript is a resubmission of an earlier submission. The following is a list of the peer review reports and author responses from that submission.

Round 1

Reviewer 1 Report

Comments and Suggestions for Authors

Comment

         In this paper, the DiffusionPoint2Polygon (DP2P) network is introduced, which models the label converter as a diffusion model, capitalizing on its noise prior and progressive refinement benefits. Specifically, we design a Multi-Scale Consistent Noise Generator that introduces multiple bags of random noise points for each point annotation and maps them onto multi-scale feature maps to complete the diffusion process. This study has certain innovative, especially in the use of noise priors and progressive refinement. The article has the following problems: 

1. The manuscript, at least in its current form, does not meet the criteria for novelty and impact expected for papers in the field. The reason is that the workload of the manuscript content is less, and the application of the formula is relatively outdated and lacks of innovation.

2. Although DP2P network is attractive in theory, the article has some shortcomings in demonstrating the significant improvement of its methods relative to existing technologies. Especially in comparison with existing technologies, there is a lack of in-depth analysis of why this method is better.

3. The conclusions drawn are do not appear to be particularly insightful or thought-provoking. In particular, the significance of the research outcomes in section 5 is limited and predictable.

4. Although the experimental part covers the DOTA and DIOR-R datasets, there is a lack of comparative experiments to show the performance changes of DP2PNet under different conditions, such as different noise levels or different types of targets.

5. It is suggested to add some theoretical support and mathematical proof to the DP2P model.

Based on the above considerations, I believe that the manuscript failed to meet the criteria for publication. It is suggested that the author make major revisions to the article, especially in the clarity of methodology, the comprehensiveness of experimental design and the depth of technology.

Reviewer 2 Report

Comments and Suggestions for Authors

1. the content of the research review is too little, especially the research content related to this paper, such as part 2.2. In order to reflect the innovation of this paper, the paper should analyze the previous related research and the unsolved problems in detail

2. Is table1 properly positioned in the text?

3. from the data in Table 1 and table 2, the advantages of the model in this paper over some other models are not obvious. There is no further explanation in this paper, nor a more in-depth analysis of the reasons, nor a comparison of other parameters between the models to illustrate the rationality of the method used in this paper

4. the fourth part is mainly the result description. As a research paper, if the method used in this paper is lack of deeper analysis, it is only the result description, which is lack of depth and innovation

Reviewer 3 Report

Comments and Suggestions for Authors

The work "DiffusionP2P: Achieving Oriented Object Detection by Generating Inspiration-Aligned Polygon with Single Point Supervision" introduces DP2PNet for modeling the point converter as a diffusion model. The paper includes a description of the main approaches of oriented object detection. Special attention is paid to fine-grained RBBs. The overall structure of the DP2PNet is presented and described. The authors adhere to a very detailed presentation, which will allow, if necessary, to reproduce the approach described in this work. The approach underlying the DP2PNet work is described in detail. All necessary references to the software solutions used are provided. The References section contains relevant works. The process of building DP2PNet and its training is described. The most well-known datasets were used for training, which simplifies comparison with known solutions. The obtained results of using DP2PNet demonstrate the high quality of the developed net. As the authors themselves note, this approach eliminates the reliance on complex prior information. These words fully reflect the main result of the work. The work leaves a positive impression, is clearly written and deserves publication in the Sensors journal after minor adjustments and responses to the following comments:

1. There is no difference between Figures 1 and 2. The authors should replace the figure with the caption with the correct one, remove the duplication, or split this figure into two.

2. Line 158 "During training, a neural network is trained to predict predict". There are also other typos in the text. This should be corrected.